# A comparison of national seasonal influenza treatment guidelines across the Asia Pacific region

Ellen Beer[1]*, Simon Boyd[1,2], Phrutsamon Wongnak[1], Thundon Ngamprasertchai[3], Nicholas J. White[1,2]

1 Mahidol Oxford Tropical Medicine Research Unit, Faculty of Tropical Medicine, Mahidol University, Bangkok, Thailand, 2 Nuffield Department of Medicine, Centre for Tropical Medicine & Global Health, University of Oxford, Oxford, United Kingdom, 3 Department of Tropical Hygiene, Faculty of Tropical Medicine, Mahidol University, Bangkok, Thailand

* ellen.m.beer@gmail.com

## Abstract

Seasonal influenza leads to 2–3 million infections and up to 650,000 global deaths annually, with particularly high mortality in Asia and relatively low annual vaccination rates for prevention. Relatively lower attention is paid to antiviral treatment as a facet of influenza response strategy both in research and national policy. This study compares national influenza treatment guidelines across countries in the Asia Pacific region, and assesses the antiviral recommendations, comprehensiveness, availability, and quality, compared with World Health Organisation (WHO) guidelines. Ministry of Health websites were searched, and key stakeholders were contacted to obtain national influenza treatment guidelines. Official guidelines detailing pharmacologic treatment for seasonal influenza were included. Key data for comparison were extracted and quality appraisal was conducted using the AGREE II instrument. Out of 49 countries and areas in the World Health Organisation Western Pacific and South-East Asia regions, under half (14/49; 28.6%) had established national influenza treatment guidelines. Nine (9/49; 18.4%) reported no seasonal flu guidelines at all, and information could not be obtained for 25 (51.0%). All guidelines recommend oseltamivir in line with WHO recommendations, although rationale and evidence reviews were often missing. There was variation in recommendations for other antivirals, indications for treatment, definitions of severity and recency of publication. The AGREE II tool quality assessments revealed the highest average scores were observed in the 'presentation' domain and lowest scores in 'editorial independence' and 'rigour of development' domains, demonstrating limited evidence-based guideline development. The variability in recommendations and definitions highlight the need for a stronger evidence base with direct comparisons of antiviral treatment for hard and soft endpoints, and improvements in systematic guideline development. Established treatment guidelines are a key component of national influenza response strategy

**Data availability statement:** All relevant data are within the paper and its Supporting Information files.

**Funding:** This study was supported in part by a Wellcome Trust grant through the COVID-19 Therapeutics Accelerator [223195/Z/21/Z to NJW]. The funders had no role in study design, data collection and analysis, decision to publish, or preparation of the manuscript.

**Competing interests:** The authors have declared that no competing interests exist.

and in the post-covid pandemic era, renewed attention to seasonal influenza management is surely warranted.

## Introduction

Seasonal influenza epidemics cause an estimated 2–3 million infections per year globally and up to 650,000 deaths [1]. Excess mortality rates are estimated to be higher in Asia than elsewhere. There is a substantial morbidity burden in children and the elderly [1–4]. While the Global Influenza Surveillance and Response System characterises circulating strains from National Influenza Centres across Asia Pacific, incidence and morbidity rates rely on extrapolations from inpatient and outpatient data, as seasonal influenza is not a notifiable disease [2].

Seasonal influenza vaccination remains the main strategy to prevent influenza morbidity and mortality. Each year globally circulating influenza virus strains are reviewed and selected by the World Health Organisation (WHO) vaccine composition committee for vaccine development. Current production timeframes mean these are distributed six months later, which carries a risk of antigenic mismatch between the vaccine and the eventual circulating virus. In the Northern Hemisphere, strain review and decisions are made in February, so that the vaccine can be distributed ahead of winter. Southern Hemisphere recommendations are made each September. Influenza vaccines are not invariably distributed equitably. National influenza vaccination campaigns are currently rolled out in half of Asia Pacific countries [5,6]. One multi-country review of studies assessing influenza vaccination studies in Asia reported a median uptake of 14.3% across general populations [7]. Notably, mRNA or self-amplifying-mRNA based vaccine candidates pose significant opportunity. They will have the advantage that one can wait longer to decide on strains as the manufacturing process is quicker, hence having potentially better matches as well as the likely inclusion of neuraminidase antigen protection. However, the current combination of limited annual vaccine coverage, the six-month long vaccine production time and margin for antigenic mismatch means that influenza remains a substantial public health threat.

Several anti-influenza drugs have been developed. Use of these antiviral drugs may reduce the risk of death, hospitalisation, complications and reduce the severity and duration of symptoms, although the evidence for these benefits is variable. There is general consensus that anti-influenza drugs can shorten the duration of illness but their effects on progression to severe disease, and on mortality remain uncertain. While antiviral drugs form a crucial component of national influenza preparedness and response plans, most clinical research is on vaccination and surveillance. Between 2014 and 2021 some 344 clinical trials were reported on influenza vaccines, and only 37 on antiviral treatments, with just eight until 2020 reporting direct drug comparisons [8,9]. This contrasts with over 4000 clinical trials recorded for covid-19 since 2020 [10]. Fig 1 illustrates registered influenza randomised control trials since 1965 on PubMed.

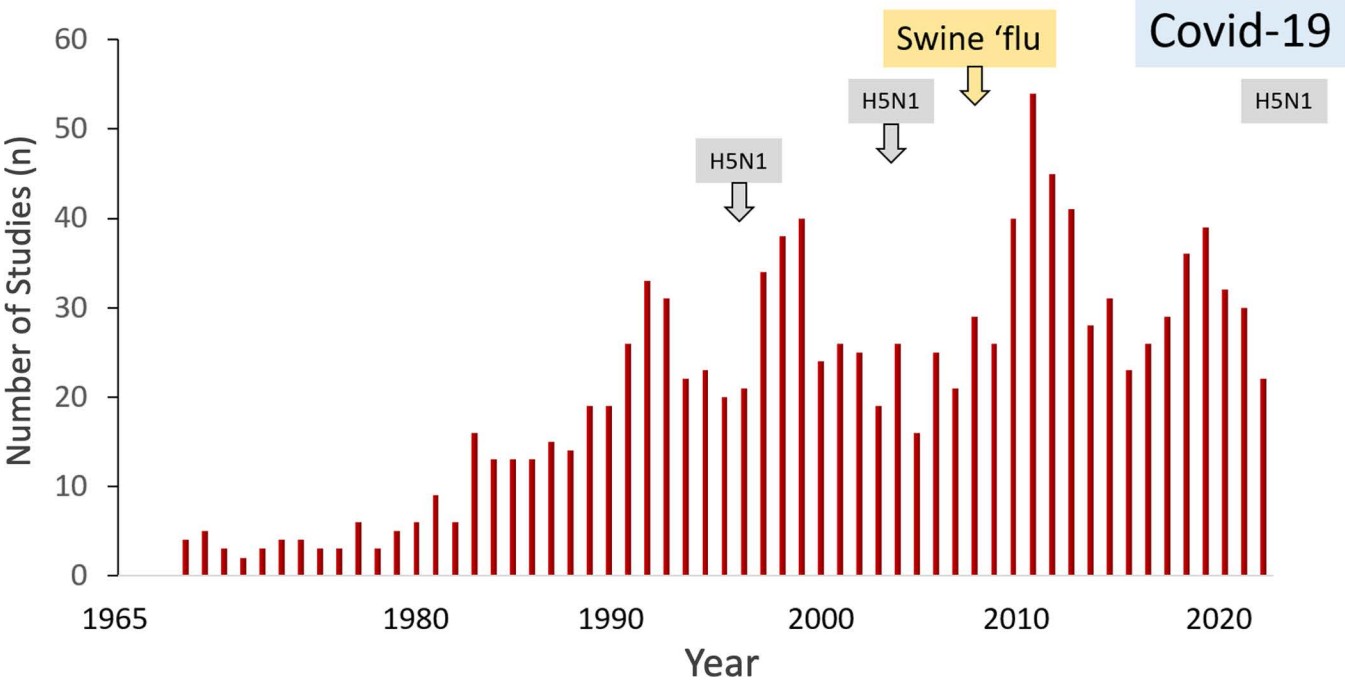

**Fig 1. Influenza treatment randomised control trials 1965 to 2024.** *PubMed search "Influenza treatment" NOT "vaccine" NOT "Covid19" filtered for randomised controlled trials. Epidemics, pandemics, and concerns for pandemics are shown in shaded boxes above.*

### Influenza antiviral drugs on the market

**M2 ion channel inhibitors.** The M2 ion channel inhibitors amantadine and rimantadine were widely recommended for many years but they have limited clinical use because of influenza A(H1N1)pdm09, A(H3N2) resistance, and lack of activity against influenza B [11,12]. The oral neuraminidase inhibitor (NAI) oseltamivir has since superseded M2 inhibitors as first line treatment against influenza A and B, with inhaled zanamivir as an alternative. There are now several other NAIs available and effective drugs with other mechanisms of action (favipiravir, baloxavir).

**Oseltamivir.** Oseltamivir phosphate was approved by the United States Food and Drug Administration for the treatment of influenza in adults in 1999 [13]. Oseltamivir is considered safe for use in pregnancy and children. Prospective, randomised placebo-controlled trial data indicates that treatment of adult influenza with oseltamivir reduces symptom severity and duration by approximately 24 hours in healthy individuals with acute uncomplicated influenza [14–17]. The 2014 Cochrane systematic review found oseltamivir reduced time to first alleviation of symptoms by 16·8 hours (95% Confidence Interval (CI); 21·8 to 8·4), in intention-to-treat adult populations with influenza-like illness and a reduction by a mean difference of 29 hours (95% CI: 12–47 hours; $P = 0.001$) in influenza infections in otherwise healthy children. The more recent ALIC$^4$E multi-centre randomised controlled trial (RCT) showed a mean reduction in symptoms by 1·02 days (95% CI 0·74–1·31) in outpatients presenting with influenza-like illness overall, and by 3·20 (95% CI 1·00–5·50) in patients aged 65 years or older who had more severe illness, comorbidities, and longer previous illness duration [18]. The effect of oral oseltamivir on mortality has been assessed only in observational studies. Randomised controlled clinical trials to date have been insufficiently powered to assess the impact of oseltamivir on mortality. Pooled odds ratios (ORs) suggest oseltamivir reduces mortality risk by a third; 0.38 (95% confidence interval, CI, 0.19–0.75), compared to placebo in all-age high-risk patients in eight observational studies [19]. The studies have assessed influenza A H3N2, H5N1, and pandemic H1N1 strains. These observational flu studies are considered not to be affected by survivorship

bias [20]. Oseltamivir given within 48 hours of symptom onset was associated with reduced mortality in 2124 patients with influenza pneumonia admitted to ICU when compared to later treatment (OR 0.69, 95% CI 0.51–0.95). However despite adjustments this may be a biased estimate [21]. A 2015 meta-analysis of oseltamivir placebo-controlled trials noted slightly lower rates of lower respiratory tract complications requiring antibiotics; risk difference -3·8% (95% CI −5·0 to −2·2), hospitalisation (risk difference -1·1%, 95% CI -1·4 to – 0·3) [22].

Oseltamivir has been included in the WHO Essential Medicines List since 2011, for conditional use in severe and or hospitalised patients with confirmed or suspected influenza [23]. It has remained in each update since its inclusion, but has been moved to the complementary list in 2017 which reflects less favourable cost-effectiveness. Oseltamivir resistance resulting from point mutations in the neuraminidase gene, (most commonly the H275Y mutation), can be selected (particularly in high viral load infections such as H5N1 pneumonia) and occur naturally in some strains. Dominant strains such as A(H1N1)pdm09, continue to show low levels (~1–2%) of resistance [24].

**Zanamivir.** Inhaled zanamivir is administered twice daily for five days, and its efficacy for soft outcomes is comparable to oseltamivir in reducing symptom duration [25]. It is safe in pregnancy. However few intervention studies directly compare both treatments for critical outcomes. An intravenous formulation is available for severe influenza.

**Peramivir.** Peramivir can be given by intravenous or intramuscular injection. Peramivir significantly reduced viral titres on day three and reduced symptom duration compared to placebo in an RCT conducted in uncomplicated influenza [26,27]. It was non-inferior to oseltamivir in reducing symptom duration [27]. Data are insufficient for influenza B or complicated cases.

**Laninamivir.** Inhaled laninamivir, currently licensed only in Japan, is administered in a single inhalation. It has been shown non-inferior to oseltamivir in the interval to illness resolution in adults with uncomplicated influenza [28].

**Baloxavir.** The cap-dependent endonuclease inhibitor baloxavir marboxil became available in 2018 with subsequent studies concentrated in Japan. It is relatively slowly eliminated (half-life 50–90 hours) and so is given as a single oral dose. It is the most expensive option at present for the treatment of uncomplicated influenza. Oral baloxavir has shown a greater early reduction in viral titres compared with oseltamivir but similar efficacy to oseltamivir in ameliorating symptoms in adolescents and adults with uncomplicated influenza [29,30]. Studies have shown consistent seasonal influenza subtypes and lineages are susceptible to baloxavir [31]. Due to lack of data, it is not recommended currently for use in pregnancy.

**Favipiravir.** The RNA polymerase inhibitor favipiravir has demonstrated significant reductions in viral titres and shortened duration of viral shedding in two phase three randomised placebo-controlled trials in uncomplicated influenza, but the evidence on time to symptom alleviation has been inconsistent [32]. Animal studies support efficacy against HPAI A(H5N1) and high barrier to resistance, but teratogenicity observed in animal studies prevents use in pregnancy.

**Umifenovir.** Umifenovir is an antiviral drug used mainly in Russia and China. It inhibits fusion between the viral envelope and the host cell membrane preventing viral entry to the target cell, and therefore protecting it from infection [33]. The clinical efficacy is uncertain.

**Other.** Treatment with corticosteroids is associated with increased odds of mortality and hospital acquired infection rather than providing benefit but the 21 observational studies from which this has been concluded may be subject to confounding by indication [34].

## Objectives

To date, WHO country influenza statements do not include information on national influenza treatment strategies or guidelines, and no publicly available papers have carried out a review of influenza treatment guidelines at the national level across countries in the Asia Pacific region.

Our objectives therefore were to investigate the availability, comprehensiveness, and antiviral recommendations across national influenza treatment guidelines in Asia Pacific countries, and to compare these recommendations with 2021 WHO guidelines.

## Methods

We conducted a review of national seasonal influenza treatment guidelines for countries in the WHO South-East Asia (SEARO) and Western Pacific Region (WPRO). National clinical guidelines were searched for and obtained between June and September 2024 using the following stepwise approach, adapted from Cokljat et al. [35].

1. Ministry of Health and National Communicable Disease Department or Society websites were searched, using google translation tools where necessary to identify guidelines.

2. Where guidelines were not readily available, obvious national level communicable disease contacts were identified from the website and contacted via email.

3. If steps 1 and 2 were unsuccessful key topic researchers in the respective countries were identified via key guideline or policy papers published locally and emailed with a maximum of a single follow up.

4 .If steps 1, 2 and 3 were unsuccessful WHO country representative office communicable disease focal points were contacted with a maximum of a single follow up.

5. Where identified guidelines were not clearly marked as a recognised national guideline, researchers identified by steps 2 and 3 were contacted for confirmation.

6. If key topic researchers responded to state that there is no official national seasonal influenza guideline in place, this was recorded as 'no guideline in place'. When a response was not received, guideline status was recorded as 'unknown', it cannot be inferred that a guideline is not in place, rather than publicly inaccessible.

### Inclusion and exclusion criteria

National guidelines for countries and areas within SEARO and WPRO regions were assessed for inclusion in this review. Guidelines were included if they clearly outlined the clinical management of influenza including therapeutic treatment and were confirmed to be the official national guideline by an official ministry of health website, national communicable disease agency or key author. Though guidelines ranged from applying only to seasonal influenza through to management of all types, this review focussed on and comparing seasonal influenza guidelines. The most up-to-date version was included. This assumed that versions obtained from national websites were the most up to date versions. Guidelines that were sub-national or facility-based, unofficial, vaccination- or surveillance-only were excluded. Guidelines were excluded if no clinical management detail was included beyond stating a named antiviral drug, for example public health unit targeted or infection prevention and control orientated guidelines. No language exclusions were made. Prophylaxis, antibiotic use for secondary infection, and supportive therapy was not considered. Whether recommended drugs were actually available could not be assessed.

### Data extraction

Two authors carried out the guideline search, selection and data extraction. A third investigator was available to resolve discrepancies. The following data points were extracted: publishing institution, year published, definition of severity, diagnostic methods, indications for testing and treating, recommended antiviral(s), dose information, order of recommendation, safety profile information, regulatory/licensing status of the recommended antiviral. Detailed extraction criteria can be found in the table in S1 Table. Google translation tools were used to extract data from guidelines that were not in the authors' spoken languages.

### Quality appraisal

Two reviewers independently appraised each guideline using the Appraisal of Guidelines for Research and Evaluation II (AGREE II) instrument. The AGREE II is a standardised quality and comprehensiveness assessment tool comprised of

six domains: scope and purpose, stakeholder involvement, rigour of development, clarity of presentation, applicability and editorial independence [36]. Each reviewer utilised the AGREE II manual to understand the process, and scored each domain on a 7-point Likert scale. After all guidelines were assessed, a review was performed to identify any discrepancies in scores of 4 or greater between appraisers. These items were discussed to reach a closer consensus before analysis. A senior author was available for consultation to resolve any persistent discrepancies in scoring – this was not found to be necessary. Domain totals were converted to a percentage of the maximum possible score for each domain and number of reviewers, according to the tool instructions.

## Results

### Availability

Of the 49 countries and areas in the WPRO and SEARO regions, established national guidelines were confirmed to be in place in 14 (14/49; 28.6%): Australia, China, Hong Kong, India, Japan, Kiribati, Malaysia, Myanmar, Nepal, Taiwan, South Korea, Sri Lanka, Thailand and Vietnam. These were denoted by 20 documents in total that were included in the analysis (Fig 2). Nine countries (9/49; 18.4%) confirmed that there were no seasonal flu guidelines in place: Bangladesh, Cambodia, Laos, Indonesia, New Zealand, Palau, Philippines, Singapore and Vanuatu. Bangladesh and Palau confirmed the 2021 WHO guidelines are the officially recommended reference for practitioners. They have not adapted these into a national version. We do not include Bangladesh or Palau in analysis of national guidelines for this report given the objective to compare national versions with WHO guidelines. Guidelines could not be retrieved through search methods or via contacting key policymakers and researchers from 25 countries (25/49; 51.0%): American Samoa, Bhutan, Brunei, Cook Islands, Democratic People's Republic of Korea (DPRK), Fiji, French Polynesia, Guam, Macau, Maldives, Marshall Islands, Federated States of Micronesia, Mongolia, Nauru, New Caledonia, Niue, Northern Mariana Islands, Papua New Guinea, Samoa, Solomon Islands, Timor Leste, Tokelau, Tonga, Tuvalu and Wallis and Futuna. The publication years for the seasonal influenza guidelines ranged from 2011 to 2024, with a median publication year of 2019. The majority (11/14; 78.6%) were published prior to the most recent WHO guideline in 2021. Of the 14 national guidelines, seven (7/14; 50.0%) specifically addressed seasonal influenza; India, Japan, Myanmar, South Korea, Sri Lanka, Taiwan and Vietnam, while six; China, Hong Kong, Kiribati, Malaysia, Nepal and Thailand (6/14; 42.9%), along with the WHO guideline at the time of review, applied to all types of influenza. The Australian guidelines applied to seasonal and zoonotic influenza.

### Settings

The WHO, Japan, South Korea, Taiwan, India, and Vietnam guidelines were applicable for use at all levels of the health system. Guidelines from Hong Kong, Malaysia, Kiribati, Sri Lanka, and Myanmar were for outpatient settings, attached to or separate to hospitals.

### Diagnostics

Recommendations across all national guidelines did not restrict the initiation of treatment to confirmed cases. WHO pragmatically recommends diagnostic testing by batch reverse transcriptase polymerase chain reaction (RT-PCR) or other rapid molecular influenza assays when a result is available within 24 hours. When a rapid result is not available, it is recommended to start treatment and then re-evaluate with the result or treat empirically when testing is unavailable. Guidance was aligned in China, Nepal, Taiwan, South Korea. Testing was limited to severe presentations in India, Myanmar, or in hospitalised patients in Sri Lanka, Australia, Japan, Malaysia. Vietnam discuss diagnostic methods but do not indicate whether confirmation of influenza should precede treatment in indicated groups. South Korean guidelines do not recommend testing during epidemics. Kiribati do not recommend testing due to unavailable resources.

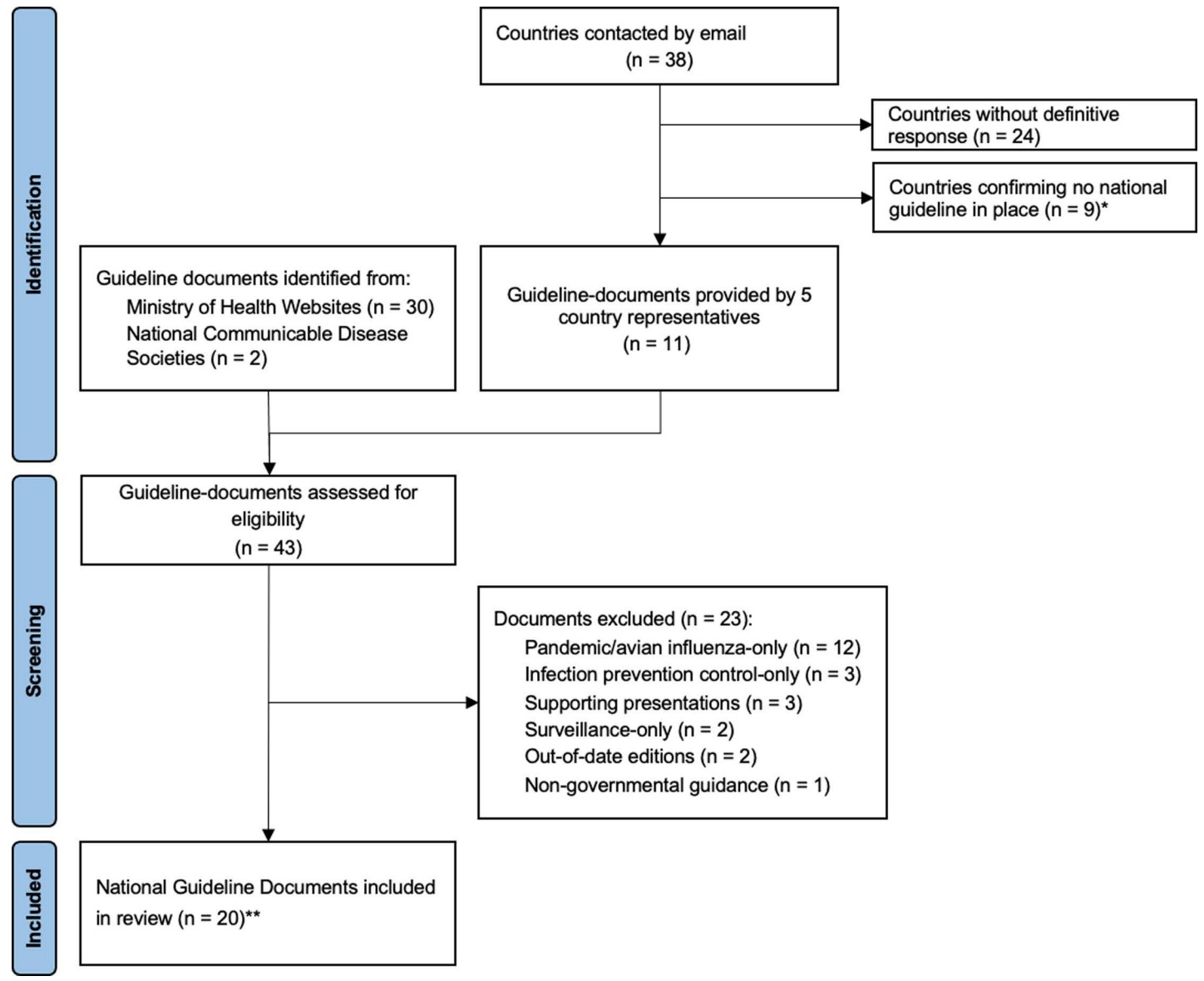

**Fig 2. Flow chart demonstrating the retrieval of national guidelines for 14 countries.** *Guidelines comprised of 20 individual documents for 14 countries. *Bangladesh and Palau confirmed no national adaptation of guidelines in place but WHO guidelines are the official guideline practitioners are advised to refer to. **Multiple separate documents formed the final guideline document set in each of India and Myanmar – treatment flow charts were arranged in separate document files in these two countries.*

## Indication

Antiviral treatment was indicated in all severe cases and individuals in high-risk groups. The majority of guidelines (9 out of 14; 64.3%) advised treatment of individuals with severe influenza illness requiring hospitalisation. The broadest indications for treatment came from China, Japan, South Korea, and Thailand where uncomplicated suspected or confirmed flu presenting within 48 hours of symptom onset 'may be considered for treatment'. Treatment of influenza in healthcare workers, particularly those working with high-risk patients, was recommended by WHO, Japan and South Korea. Additionally, Japan, South Korea and Australia also highlight treatment of patients with mild illness who have household contacts

in high-risk groups. Table 1 breaks this down by country. The timeframe for treatment of indicated groups is 'as soon as possible' by WHO. This is echoed by all 14 guidelines. Ten out of 14 (10/14; 71.4%) recommend aiming for treatment within 48 hours of symptom onset, but do not restrict treatment to this window.

### Definitions for severe influenza

Definitions for severe influenza varied and were often imprecise or ambiguous. Twelve guidelines provided a definition for severe influenza. Two guidelines did not (Japan and Hong Kong). WHO defined severe cases as those with "*illness that would lead to hospitalisation, including development of clinical syndromes: severe pneumonia or sepsis; or exacerbation of underlying chronic diseases*". It describes uncomplicated influenza as *illness characterized by a sudden onset of cough, headache, muscle and joint pain, severe malaise, sore throat, and a runny nose, with or without fever*.

Interestingly, most guidelines (9/12; 75.0%) used an influenza like illness case definition for uncomplicated flu, characterised by a fever ≥38 plus other variable symptoms. For the severe definition only three guidelines used the word 'hospitalisation' however most (8/12) guidelines indicated that if patients exhibit complications such as sepsis or ARDS that would likely require hospitalisation, this is considered severe. Australian therapeutic guidelines defined severe cases more narrowly, confined only to those requiring "*haemodynamic or ventilatory support*". Two guidelines include 'other reasons for hospitalisation' as an independent marker for severe illness (China, South Korea), open to interpretation. Ten guidelines describe signs or symptoms of lower respiratory tract infection, with or without radiological evidence as part of their definition. Seven guidelines highlight the deterioration of underlying co-morbidities as a characteristic of severe illness in alignment with the WHO definition. Three guidelines (Malaysia, China, Taiwan, and India) use 'persistence of symptoms' or 'high fever for more than 72 hours' as a marker of severe illness. One guideline included diarrhoea as an independent marker of severe illness. Two guidelines specified objective oxygen saturation thresholds for severe illness: < 90% and < 95%. Some descriptions of markers of severity were open to interpretation such as "breathlessness", "fast breathing", "chest pain", "somnolence", "decreased urine output", "watery diarrhoea", "serious symptoms".

### High-risk groups

All 15 guidelines characterised groups at high risk of complications. All 14 countries included pregnant and post-partum individuals. Children defined as < 5 years were high risk in 10/14 (71.4%) guidelines. Nepal narrowed this to children who are malnourished. Of these 10, seven also specified infants. Sri Lanka, Thailand, Taiwan, and Hong Kong considered infants but not children to be high risk. Taiwanese guidelines do not specify whether antivirals are specifically confined to high-risk groups for this to affect treatment decisions. Neither children nor infants are considered high risk in Myanmar guidelines. Eleven countries considered older individuals: over 65 or over 60. People younger than 18–19 years of age on long-term aspirin- or salicylate-containing medications were highlighted as being at increased risk of complications (Reye's syndrome) in five guidelines. A BMI over 40 was included as a "high risk" factor in WHO and 10 guidelines. Taiwan, Australia, Thailand, and China extended the group to a BMI over 30 kg/m$^2$, while Vietnam limited the 'obesity' severity criterion to children. Myanmar guidelines did not include obesity. All 14 countries included patients with co-morbidities as being at high risk, with some variation in level of detail. Similarly, immunocompromise or HIV positivity were highlighted by all, in alignment with WHO. Beyond groups highlighted by WHO, Australia, Japan and South Korean guidelines also include nursing home residents. Australian guidelines also include houseless patients, and Aboriginal and Torres Strait Islander people of any age.

### Recommended antiviral(s)

WHO recommends oseltamivir in indicated groups and discourages use of any other NAI or M2 inhibitor for the treatment of influenza. Baloxavir and favipiravir were not reviewed in the most recent update. All 14 countries with a national

**Table 1. Treatment indications for suspected or confirmed influenza by country-level guideline across the Asia Pacific region.**

| | National Guideline in Place | Treatment Indications<br>Any patient with confirmed or suspected influenza that is/has: | | | | | |
| --- | --- | --- | --- | --- | --- | --- | --- |
| | | Severe or compli-cated | Required hospitalisa-tion | High risk of com-plications from Influenza | Uncomplicated/low risk if treatment can be admin-istered within 48 hours of symptom onset | Household con-tacts in high-risk group | Healthcare worker |
| WHO | NA | ● | ● | ● | | | ● |
| American Samoa | Unknown | | | | | | |
| Australia | Yes | ● | ● | ● | | ● | |
| Bangladesh[a] | No | ● | ● | ● | | | ● |
| Bhutan | Unknown | | | | | | |
| Brunei | Unknown | | | | | | |
| Cambodia | No | | | | | | |
| China | Yes | ● | ● | ● | ● | | |
| Cook Islands | Unknown | | | | | | |
| DPRK | Unknown | | | | | | |
| FSM | Unknown | | | | | | |
| Fiji | Unknown | | | | | | |
| French Polynesia | Unknown | | | | | | |
| Guam | Unknown | | | | | | |
| Hong Kong | Yes | ● | | ● | | | |
| India | Yes | ● | | ● | | | |
| Indonesia | No | | | | | | |
| Japan | Yes | ● | ● | ● | ● | ● | ● |
| Kiribati | Yes | ● | ● | ● | | | |
| Laos | No | | | | | | |
| Macau | Unknown | | | | | | |
| Malaysia | Yes | b | b | ● | | | |
| Maldives | Unknown | | | | | | |
| Marshall Islands | Unknown | | | | | | |
| Mongolia | Unknown | | | | | | |
| Myanmar | Yes | ● | ● | ● | | | |
| Nauru | Unknown | | | | | | |
| Nepal | Yes | ● | ● | ● | | | |
| New Caledonia | Unknown | | | | | | |
| New Zealand | No | | | | | | |
| Niue | Unknown | | | | | | |
| CNMI | | | | | | | |
| Palau[c] | No | ● | ● | ● | | | ● |
| PNG | Unknown | | | | | | |
| Philippines | No | | | | | | |
| Pitcairn Islands | Unknown | | | | | | |
| Samoa | Unknown | | | | | | |
| Singapore | No | | | | | | |
| Solomon Islands | Unknown | | | | | | |
| South Korea | Yes | ● | ● | ● | ● | ● | ● |
| Sri Lanka | Yes | ● | ● | ●[d] | | | |

*(Continued)*

**Table 1.** (Continued)

| | National Guideline in Place | Treatment Indications<br>Any patient with confirmed or suspected influenza that is/has: | | | | | |
| --- | --- | --- | --- | --- | --- | --- | --- |
| | | Severe or compli-cated | Required hospitalisa-tion | High risk of com-plications from Influenza | Uncomplicated/low risk if treatment can be admin-istered within 48 hours of symptom onset | Household con-tacts in high-risk group | Healthcare worker |
| Taiwan | Yes | | | | | | |
| Thailand | Yes | ● | | ● | ● | | |
| Timor Leste | Unknown | | | | | | |
| Tokelau | Unknown | | | | | | |
| Tonga | Unknown | | | | | | |
| Tuvalu | Unknown | | | | | | |
| Vanuatu | No | | | | | | |
| Vietnam | Yes | ● | | ● | | | |
| Wallis and Futuna | Unknown | | | | | | |

*Abbreviations: CNMI: the Commonwealth of the Northern Mariana Islands, DPRK: Democratic People's Republic of Korea, FSM: Federated States of Micronesia; N. Mariana Islands: Northern Mariana Islands Commonwealth of the USA, PNG: Papua New Guinea, WHO: World Health Organisation.*

*\*Conditional Recommendation.*

*ª Bangladesh uses the WHO guideline.*

*ᵇ Not applicable, guidelines for outpatient primary care level only. If severe, recommends hospitalisation*

*ᶜ Palau uses the WHO guideline.*

*ᵈ High risk groups only on a 'case by case basis' – based on clinical judgement in children aged 13 and above for influenza A.*

guideline specified oseltamivir. Six (6/14, 42.9%) recommended oseltamivir alone, while nine (9/14; 64.3%) recommended at least one other antiviral. Countries recommended a median of two (range: 1–5) antivirals. Only one (Thailand) stated an order of treatment recommendation. Table 2 outlines antiviral recommendations by country, and Fig 3 illustrates the number of countries recommending each antiviral. Four guidelines discouraged the use of certain antivirals such as M2 inhibitors or favipiravir, and 12 guidelines did not make any comment on avoiding certain antivirals. Some conditions were applied, e.g., Japanese guidelines state favipiravir is reserved for novel or re-emerging strains of influenza rather than seasonal, and M2 inhibitors were only recommended conditionally based on evidence of sensitivity to the circulating strain. WHO, India, Myanmar and Australian guidelines explicitly advised against corticosteroid use. The remaining guidelines did not mention steroids.

## Dose

All guidelines apart from Myanmar and Nepal recommended the WHO recommended adult and paediatric doses of oral oseltamivir. Myanmar guidelines recommended and included treatment for adults only. Sri Lankan guidelines suggest use of oseltamivir 150 mg twice daily for 10 days in patients not responding to standard dosing. Thai guidelines advise against using a double dose but state a course longer than five days may be considered. Indian guidelines also sug-gest clinicians may extend the course at the same dose in immunocompromised individuals or patients with pneumonia. Australian guidelines acknowledge the lack of evidence but state 'a longer duration of neuraminidase inhibitor therapy can be considered in patients with severe influenza or who are immunocompromised'. Kiribati guidelines recommend that non-responding cases may be considered for a higher dose and longer duration of the oseltamivir regimen, e.g., in adults: 150 mg twice a day after discussion with clinician. No guideline recommended a loading dose in severe influenza. Chinese guidelines advise against double dosing of NAI's.

Table 2. Recommended antiviral(s) by country.

| Country or WHO | Recommended Antiviral | | | | | | | | |
|---|---|---|---|---|---|---|---|---|---|
| | Oseltamivir | Zanamivir | Peramivir | Laninamivir | Baloxavir | Favipiravir | Amantadine | Umifenovir | Herbal |
| WHO | • | | | | | | | | |
| Australia | • | • | | | | | | | |
| Bangladesh[a] | • | | | | | | | | |
| China | • | • | • | | | | | • | • |
| Hong Kong | • | | | | | | | | |
| India | • | • | | | | | | | |
| Japan | • | • | • | • | •[b] | | | | |
| Kiribati | • | | | | | | | | |
| Korea | • | • | • | | | | • | | |
| Malaysia | • | | | | | | | | |
| Myanmar | • | | | | | | | | |
| Nepal | • | | | | | | | | |
| Sri Lanka | • | | | | | | | | |
| Taiwan | • | • | • | | | | | | |
| Thailand | • | | | | | • | | | |
| Vietnam | • | • | | | | | | | |

Abbreviations: WHO; World Health Organisation.

[a] Bangladesh uses WHO guidelines.

[b] Reports there is insufficient data to support recommendation in 12–19-year-olds, administer carefully to children under 12, not recommended for immunocompromised or critically ill patients.

Eight out of 14 guidelines (8/14; 57.1%) include information about the safety profile of the antivirals including listing possible side effects or cautions in renal impairment.

## Regulatory status

Just 5/14 (35.7%) guidelines mentioned the national regulatory status for at least one of the antivirals recommended.

## Quality

Quality and comprehensiveness of national guidelines varied. This is represented by the results of the AGREE II tool in Figs 4 and 5. Countries scored well on average for their 'Clarity of Presentation': 74.8% (range 47.2-100.0%). Key antiviral recommendations were largely easily identifiable, although some guidelines included ambiguous descriptions of the target populations and severity as described.

The 'scope and Purpose' domain scored on average 55.8% (range 22.2-97.2%). Half of guidelines clearly listed objectives or stated the clinical questions and intended clinical outcomes from antivirals. Target patient populations for treatment were largely clearly defined. The definitions for "severe influenza" varied and were often imprecise or ambiguous.

'Stakeholder Involvement' scored 23.0% on average (range 2.8-66.7%). Guidelines occasionally listed the authors, and their expertise. Nine out of 14 (9/14; 64.3%) guidelines indicate the healthcare professional group(s) the guideline was intended for. Eleven out of 14 countries stated the healthcare setting the guideline was intended for. No guideline (including WHO) sought the views and preferences of the public and patient population in their national flu guideline formulation.

'Rigour of Development' scored 16.5% (1.0-80.2%), reflecting rare documentation of evidence review or discussion of rationale for the choice of antiviral. Just 5/14 (35.7%) country guidelines include at least some mention of guideline formulation methods, while 4/14 country guidelines made some effort to review the evidence to discuss the context of the

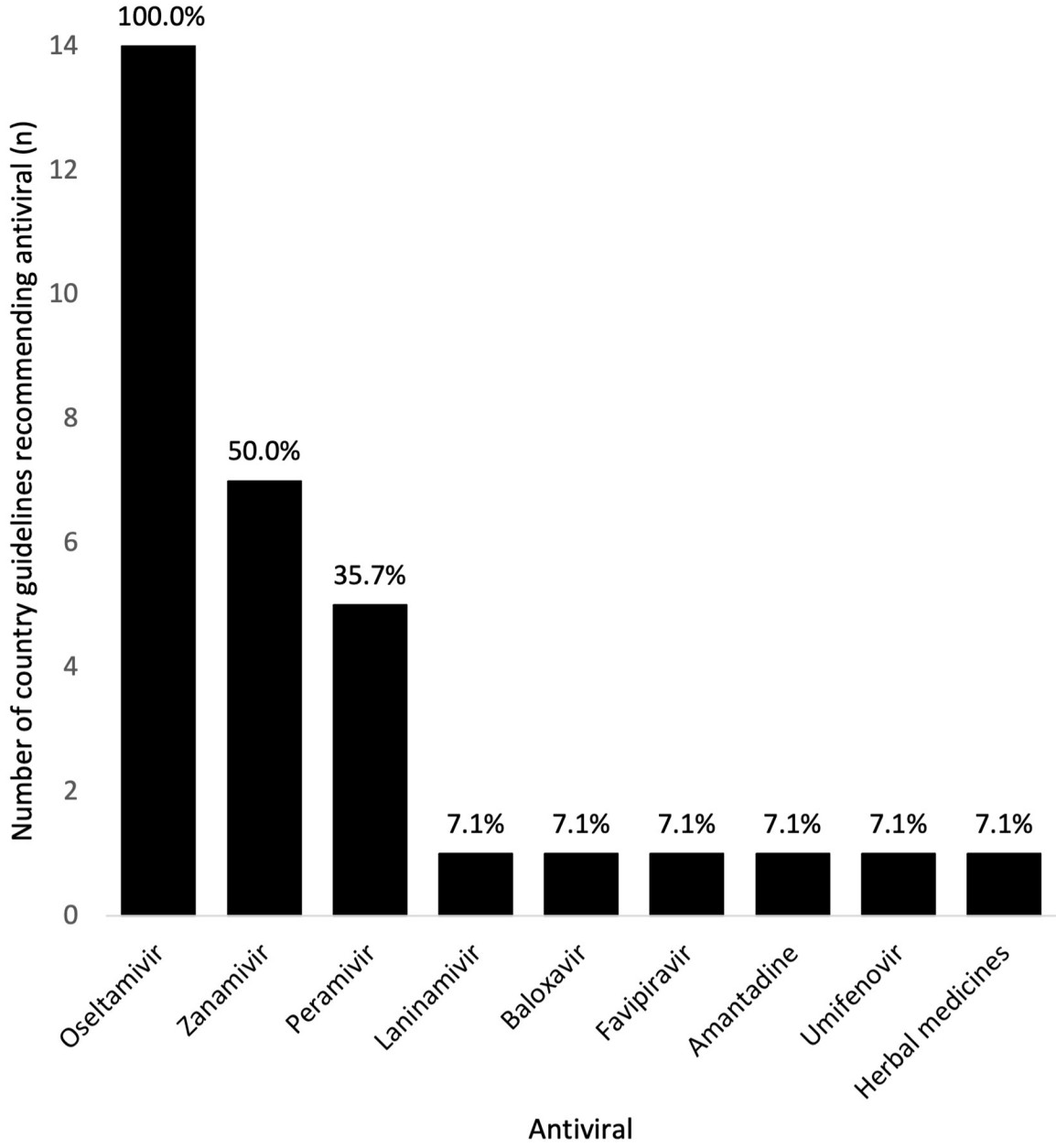

**Fig 3. Anti-influenza drugs recommended in national guidelines.**

recommendations: Australia, Japan, Kiribati, South Korea, and Taiwan. South Korea was the only guideline apart from that of WHO which applied grading to the strength of the recommendations.

'Applicability' scored 16.4% (range 2.1-41.7%). Guidelines made some effort to instruct how to facilitate implementation. None of the guidelines included a cost effectiveness analysis or discussion. Kiribati guided practitioners on alternative approaches to assessment where resources such as chest x-ray were less available, such as in the outer islands.

Guidelines scored least on 'Editorial Independence': mean 5.4% (range 0.0-54.2%). Just Japan and South Korea declared conflicts of interest or indicated the guideline had been through external review.

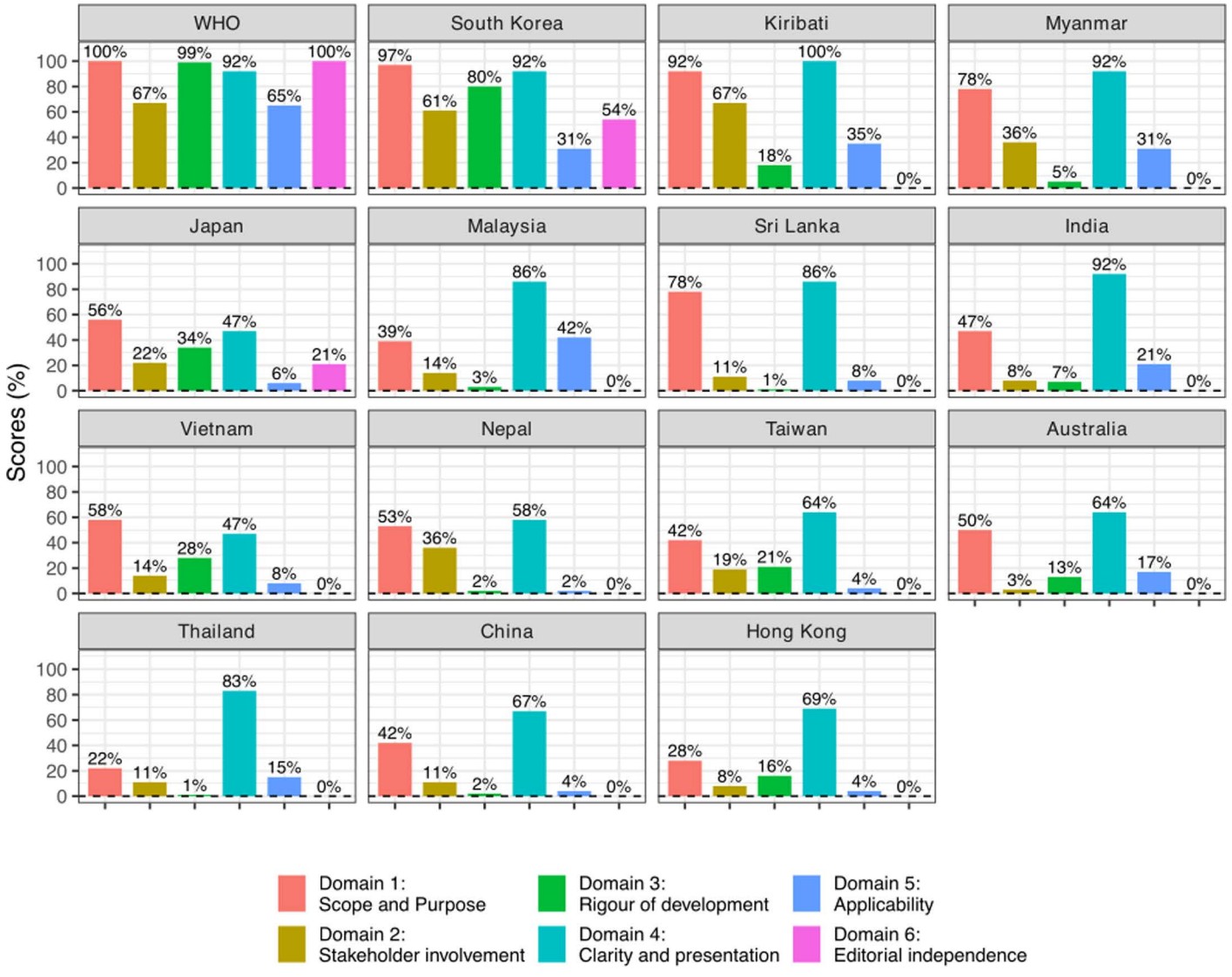

**Fig 4. Total AGREE II scores by domain for WHO and 14 national guidelines, shown as percentages of maximum possible score.**

## Discussion

Despite the large annual burden of seasonal influenza in the Asia Pacific region, the availability and comprehensiveness of national seasonal treatment guidelines remains limited. Despite the renewed global sense of urgency to address viruses of pandemic potential since the Covid-19 pandemic, and the likelihood that the next pandemic will be influenza, it is somewhat surprising that national guidelines for the management of seasonal influenza are often absent or out of date.

Clinical guideline development should be a collaborative multi-stakeholder process involving an evaluation of up-to-date therapeutic evidence, using methods such as the 'GRADE approach' to produce recommendations for practice. Recommendations should be applicable to the respective national context. As well as alignment with domestic antiviral manufacturing capacity or feasible import, distribution and storage, any limitations in human resources, and the regulatory status of each antiviral should be considered. Recommended antivirals should be a) domestically licensed b) available to practitioners and c) accessible to patients, for guideline-stated indications. Consideration of cost barriers at all levels of guideline

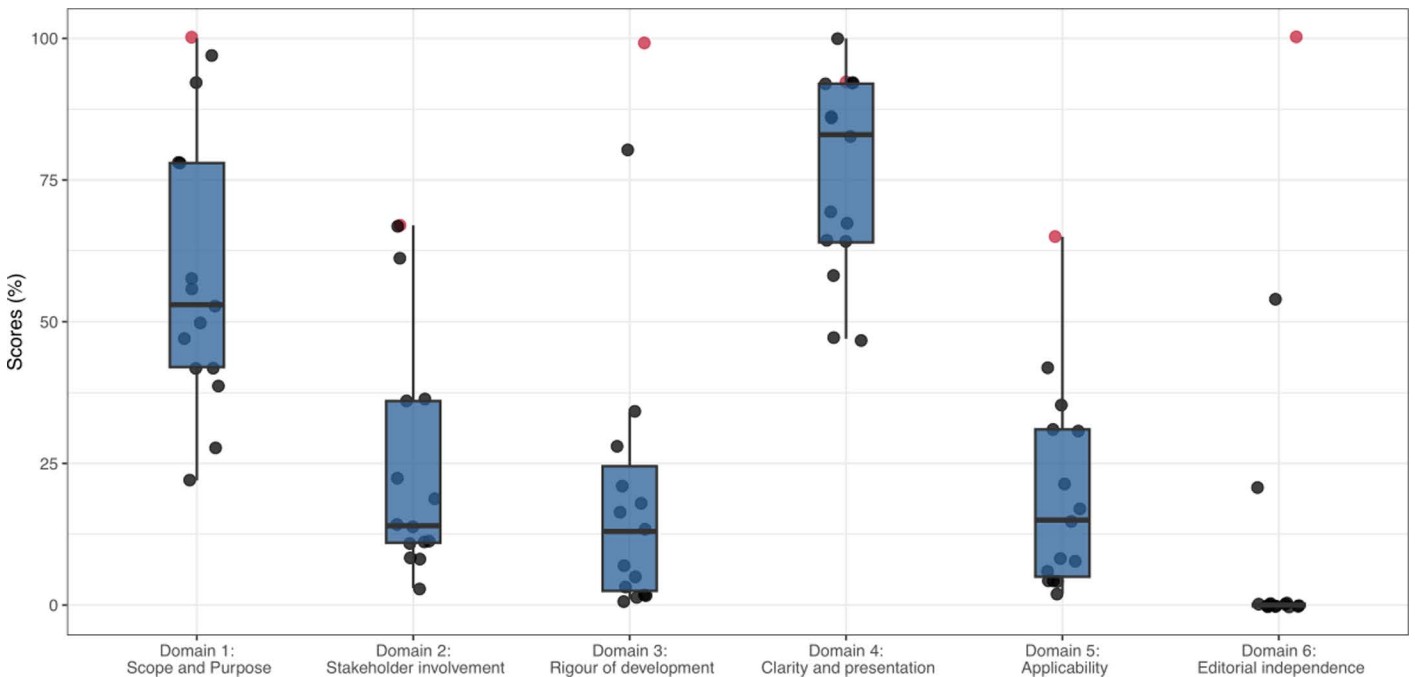

**Fig 5. Distribution of AGREE II scores by domain for WHO and 14 national guidelines.** *Red data points denote WHO guidelines.*

implementation is required to for a guideline to be useful and effective in practice. Transparent statements declaring editorial independence and any conflicts of interest indicate a guideline's dedication to evidence over any personal gains regarding antiviral choice. Such elements are included in the AGREE II evaluation tool and should inform the development process.

In our review, less than half of countries in the Asia Pacific region had an accessible national influenza treatment guideline. Most available guidelines preceded the 2021 WHO edition. Where guidelines are available, there was wide variation in guideline comprehensiveness. Eleven guidelines did not include discussion of the evidence and rationale for treatment recommendations. The level of clinical management detail was often limited with inconsistent inclusion of detailed antiviral side effects and toxicity profiles. This would be especially for any target users who are not clinicians. Simple measures such as listing guideline authors and clearly stating the objectives and target users were missing.

All influenza treatment guidelines recommend oseltamivir for severe seasonal influenza. This was in alignment with WHO. It is unclear whether oseltamivir is recommended because it considered the most effective option, the most cost-effective, the most supported by evidence or the most available option. Previous interviews to obtain national communicable disease team attitudes towards the evidence base for oseltamivir for pandemic influenza have indicated some scepticism or concern [37]. Discussion about cost, availability, and accessibility was rarely included in guidelines. Oseltamivir is widely recommended for emergent zoonotic strains and huge investments in stockpiling for pandemic preparedness have been made (largely stimulated by concerns over the pandemic potential of H5N1 influenza). The use of oseltamivir for seasonal 'flu may remain favoured simply because of application of a rollover stockpiling approach – while stock is not needed for pandemic 'flu, it can be used for seasonal cases.

Beyond oseltamivir, other NAI's and newer antivirals are included in eight guidelines. In particular, the recommendation of inhaled zanamivir was common. WHO advise against its use in severe influenza to prevent critical outcomes, citing very low certainty of benefit. Uniquely, Japan recommends and heavily prescribes laninamivir and baloxavir (both discovered in

Japan) [38]. Thailand recommends favipiravir second line option in non-severe cases and China recommends Umifenovir. It is not clear whether newer and operationally simpler single dose baloxavir has benefit over oseltamivir either in severe of in uncomplicated influenza. The variability in guidelines may result in part from the paucity of evidence. There are only 11 randomised controlled trials over the past 25 years in which the individual antiviral drugs have been compared with each other. There is a need for larger more detailed comparisons and better pharmacometric data. A clearer evidence base would greatly inform WHO and country recommendations.

Finally, there are differences in the recommendations for who should receive influenza antivirals across Asia Pacific and the definitions of severe disease. High risk groups were largely in alignment. Some countries consider treatment of individuals with high-risk contacts.

This review is limited by its scope – we were unable to obtain confirmation about guideline status for 18 out of 21 Pacific Island countries and areas using our methods. An alternative approach may be considered to review guidelines across pacific islands in future. It is possible that interpretation of guidelines was limited by the accuracy of translation tools. National guidelines can be compared but these do not necessarily reflect the status of local guidelines or whether facility level guidelines may be in place. A lack of transparency may mask where decisions were underpinned by stronger rationale than what was published. National guidelines provide recommendations only, and do not indicate whether the treatments are available to and affordable by those that need them. Large health system resource analysis studies have previously shown significant variability in distribution and availability of oseltamivir and other influenza treatment resource gaps domestically especially in rural areas in South-East Asia [39,40]. Thus guidelines may be in place but provide limited utility if ill-aligned with the reality of domestic antiviral availability. Aside from the evidence base, guideline absences may reflect limitations in healthcare infrastructure to implement influenza management guidelines or technical capacity to formulate national treatment guidelines. Finally, Influenza may or may not be regarded as a priority infection in national communicable disease strategy relative to the burden of other diseases and therefore absence of guidelines reflect low prioritisation.

Since we conducted our analysis, WHO published an updated 2024 Clinical Practice Guideline for Influenza, thus creating a timely impetus to review and update domestic influenza guidelines and management strategy [41]. The WHO guideline provides an updated summary of risk factors for severe disease and include both severe and non-severe influenza chemotherapeutic management. The update includes a review of baloxavir, favipiravir and umifenovir and as such provide a strong reference point for national guideline decision making.

In conclusion, we strongly emphasise the need for Asia Pacific influenza summits to prioritise national influenza management strategy, coordinated influenza antiviral research to inform policy and to share best practices in influenza guideline development. Lastly, to discuss regional cooperation in implementation. Establishing, updating, and implementing guidelines may affect regional and domestic antiviral demand and supply. This is further emphasised if seasonal antiviral choice overlaps with pandemic antiviral choice for stockpiling. Moreover, seasonal influenza treatment and response remains a crucial component of the global health security agenda across Asia Pacific [42,43].

## Supporting information

**S1 Table. Data extraction form.**
(DOCX)

**S2 Table. Summary of national guidelines from World Health Organisation South-East Asia region countries and areas.**
(DOCX)

**S3 Table. Summary of national guidelines from World Health Organisation Western Pacific Region countries and area.**
(DOCX)

## Acknowledgments

We would like to thank Dr Cintia Cruz and Dr Piya Hanvoravongchai for sharing their insights and advice on national guideline comparisons and studying influenza policy in Asia Pacific respectively.

## Author contributions

**Conceptualization:** Nicholas J White.

**Data curation:** Ellen Beer, Simon Boyd.

**Formal analysis:** Ellen Beer, Simon Boyd, Phrutsamon Wongnak.

**Funding acquisition:** Nicholas J White.

**Investigation:** Ellen Beer, Simon Boyd.

**Methodology:** Ellen Beer.

**Software:** Phrutsamon Wongnak.

**Supervision:** Nicholas J White.

**Validation:** Nicholas J White.

**Visualization:** Ellen Beer, Phrutsamon Wongnak.

**Writing – original draft:** Ellen Beer.

**Writing – review & editing:** Ellen Beer, Simon Boyd, Phrutsamon Wongnak, Thundon Ngamprasertchai, Nicholas J White.

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
