## [Decision Letter · Decision Letter 0]

1 Dec 2024

PGPH-D-24-02436

A Comparison of National Seasonal Influenza Treatment Guidelines across the Asia Pacific Region

Dear Dr. Beer,

Thank you for submitting your manuscript to PLOS Global Public Health. After careful consideration, we feel that it has merit but does not fully meet PLOS Global Public Health’s publication criteria as it currently stands. Therefore, we invite you to submit a revised version of the manuscript that addresses the points raised during the review process.

Please note that we have only been able to secure a single reviewer to assess your manuscript. We are issuing a decision on your manuscript at this point to prevent further delays in the evaluation of your manuscript. Please be aware that the editor who handles your revised manuscript might find it necessary to invite additional reviewers to assess this work once the revised manuscript is submitted. However, we will aim to proceed on the basis of this single review if possible. 

The reviewer is positive overall about the manuscript, but requested some clarifications, and suggestions to improve the contextualization of the study. Please remember to ensure that any recommendations are clearly marked as such; any statements identified as conclusions should be fully supported by the results presented.

We look forward to receiving your revised manuscript.

Kind regards,

Marianne Clemence

Staff Editor

Journal Requirements:

Additional Editor Comments (if provided):

Reviewers' comments:

Reviewer's Responses to Questions

**Comments to the Author**

1. Does this manuscript meet PLOS Global Public Health’s publication criteria ? Is the manuscript technically sound, and do the data support the conclusions? The manuscript must describe methodologically and ethically rigorous research with conclusions that are appropriately drawn based on the data presented.

Reviewer #1: Yes

2. Has the statistical analysis been performed appropriately and rigorously?

Reviewer #1: N/A

3. Have the authors made all data underlying the findings in their manuscript fully available (please refer to the Data Availability Statement at the start of the manuscript PDF file)?

Reviewer #1: Yes

4. Is the manuscript presented in an intelligible fashion and written in standard English?

Reviewer #1: Yes

5. Review Comments to the Author

Reviewer #1: Congratulations for this relevant and well written manuscript. Few comments:

-Line 54: statement only correct for NH recommendation, not for SH. Pls add sentence for SH

Line 59: mention here or somewhere else in the manuscript just as context that the mRNA or sa-mRNA based vaccine candidates could become a game changer in Flu management. They will have the advantage that one can wait longer to decide on strains as the manufacturing process is quicker, hence having potentially better matches. Also, it seems that most of those include also Neuraminidase Ag for improved protection

107ff: check sentences

244: which 6 countries? Please list

Figure 2: contacted countries = 38. Responders 33. and the other 5?

Discussion: good, but "so what"? What do you recommend as outcome of your research? for example, one recommendation could be that WHO includes in their recommendations details on Favipiravir , Baloxavir. Another recommendation could be that WHO regional offcies calls for a Flu summit where countries can share best practices

6. PLOS authors have the option to publish the peer review history of their article (what does this mean? ). If published, this will include your full peer review and any attached files.

**Do you want your identity to be public for this peer review?** For information about this choice, including consent withdrawal, please see our Privacy Policy .

Reviewer #1: No

---

## [Decision Letter · Decision Letter 1]

16 Feb 2025

PGPH-D-24-02436R1

A Comparison of National Seasonal Influenza Treatment Guidelines across the Asia Pacific Region

Dear Dr. Beer,

Thank you for submitting your manuscript to PLOS Global Public Health. After careful consideration, we feel that it has merit but minor revisions are suggested. Therefore, we invite you to submit a revised version of the manuscript that addresses the points raised during the review process.

The revised manuscript nicely addressed all comments by the previous peer-reviewer. As the paper had only received 1 peer-review, and the journal requires 2 peer-reviews, I was asked to serve as a new academic editor for the paper and to identify additional peer-reviewers or to conduct the peer-review. I have chosen to do the latter (in addition to reviewing the previous reviews and the revisions already made), and have three minor suggestions for the study team to consider. I look forward to reviewing the revised manuscript after these final comments have been addressed. 

We look forward to receiving your revised manuscript.

Kind regards,

Sharmistha Mishra, M.D., Ph.D

Academic Editor

Journal Requirements:

Additional Editor Comments (if provided):

This is an excellent study that uses appropriate methods to provide a comprehensive review of

guidelines. The study team also nicely addressed the previous peer-reviewer comments. I have

minor suggestions which I hope will help the presentation.

1) Please clarify how it was determined that there was “no guideline in place” and what “unknown” means in Table 1. It would be helpful to first indicate approach in the methods section (and how these two elements would thus be defined, and then refer to the results as currently included in lines 238-248).

2) It was unclear what this means, is it because no guidelines were in place? “We do not include Bangladesh or Palau in analysis of national guidelines for the purpose of this report.”

3) This statement in the discussion was interesting an important. It would be helpful to provide some context and further explanation in the discussion– (a) are these elements generally discussed in influenza guidelines and should they be discussed and/or included in treatment guidelines: “Discussion about cost, availability, and accessibility was rarely included in guidelines.”; and (b) could authors elaborate on the point made in line 480 (“Some policy papers show significant variability in distribution and availability of oseltamivir and other influenza treatment resource gaps domestically especially in rural areas in Southeast Asia.”), and comment on potential considerations if/how access/availability might shape variability in guidelines or whether guidelines exist within countries?

Reviewers' comments:

Reviewer's Responses to Questions

**Comments to the Author**

1. If the authors have adequately addressed your comments raised in a previous round of review and you feel that this manuscript is now acceptable for publication, you may indicate that here to bypass the “Comments to the Author” section, enter your conflict of interest statement in the “Confidential to Editor” section, and submit your "Accept" recommendation.

Reviewer #1: All comments have been addressed

Reviewer #2: All comments have been addressed

2. Does this manuscript meet PLOS Global Public Health’s publication criteria ? Is the manuscript technically sound, and do the data support the conclusions? The manuscript must describe methodologically and ethically rigorous research with conclusions that are appropriately drawn based on the data presented.

Reviewer #1: Yes

Reviewer #2: Yes

3. Has the statistical analysis been performed appropriately and rigorously?

Reviewer #1: N/A

Reviewer #2: Yes

4. Have the authors made all data underlying the findings in their manuscript fully available (please refer to the Data Availability Statement at the start of the manuscript PDF file)?

Reviewer #1: Yes

Reviewer #2: Yes

5. Is the manuscript presented in an intelligible fashion and written in standard English?

Reviewer #1: Yes

Reviewer #2: Yes

6. Review Comments to the Author

Reviewer #1: Thanks for including the recommendations and also for highlighting the new WHO recommendation.

Reviewer #2: This is an excellent study that uses appropriate methods to provide a comprehensive review of

guidelines. The study team also nicely addressed the previous peer-reviewer comments. I have

minor suggestions which I hope will help the presentation.

1) Please clarify how it was determined that there was “no guideline in place” and what “unknown” means in Table 1. It would be helpful to first indicate approach in the methods section (and how these two elements would thus be defined, and then refer to the results as currently included in lines 238-248).

2) It was unclear what this means, is it because no guidelines were in place? “We do not include Bangladesh or Palau in analysis of national guidelines for the purpose of this report.”

3) This statement in the discussion was interesting an important. It would be helpful to provide some context and further explanation in the discussion– (a) are these elements generally discussed in influenza guidelines and should they be discussed and/or included in treatment guidelines: “Discussion about cost, availability, and accessibility was rarely included in guidelines.”; and (b) could authors elaborate on the point made in line 480 (“Some policy papers show significant variability in distribution and availability of oseltamivir and other influenza treatment resource gaps domestically especially in rural areas in Southeast Asia.”), and comment on potential considerations if/how access/availability might shape variability in guidelines or whether guidelines exist within countries?

7. PLOS authors have the option to publish the peer review history of their article (what does this mean? ). If published, this will include your full peer review and any attached files.

**Do you want your identity to be public for this peer review?** For information about this choice, including consent withdrawal, please see our Privacy Policy .

Reviewer #1: No

Reviewer #2: **Yes: ** Sharmistha Mishra

---

## [Editor Report · Decision Letter 2]

14 Mar 2025

A Comparison of National Seasonal Influenza Treatment Guidelines across the Asia Pacific Region

PGPH-D-24-02436R2

Dear Dr Beer,

We are pleased to inform you that your manuscript 'A Comparison of National Seasonal Influenza Treatment Guidelines across the Asia Pacific Region' has been provisionally accepted for publication in PLOS Global Public Health.

Best regards,

Sharmistha Mishra, M.D., Ph.D

Academic Editor

All comments/suggestions have been addressed.